# Matrix Metalloproteinase-3 is Key Effector of TNF-α-Induced Collagen Degradation in Skin

**DOI:** 10.3390/ijms20205234

**Published:** 2019-10-22

**Authors:** Ursula Mirastschijski, Blaž Lupše, Kathrin Maedler, Bhavishya Sarma, Arlo Radtke, Gazanfer Belge, Martina Dorsch, Dirk Wedekind, Lisa J. McCawley, Gabriele Boehm, Ulrich Zier, Kazuhiro Yamamoto, Sørge Kelm, Magnus S. Ågren

**Affiliations:** 1Center for Biomolecular Interactions Bremen, Department of Biology and Biochemistry, University of Bremen, 28359 Bremen, Germany; mirastsc@uni-bremen.de (U.M.); blazpikalupse@gmail.com (B.L.); kmaedler@uni-bremen.de (K.M.); bhavishyasarma@yahoo.co.in (B.S.); u.x.zier@gmail.com (U.Z.); skelm@uni-bremen.de (S.K.); 2Faculty of Biology and Biochemistry, University of Bremen, 28359 Bremen, Germany; arlo.radtke@uni-bremen.de (A.R.); belge@uni-bremen.de (G.B.); 3Institute for Laboratory Animal Science, Hannover Medical School, 30625 Hannover, Germany; Dorsch.Martina@mh-hannover.de (M.D.); Wedekind.Dirk@mh-hannover.de (D.W.); 4Department of Biomedical Engineering, Vanderbilt University, Nashville, TN 37232-6840, USA; lisa.mccawley@Vanderbilt.edu; 5Department of General, Visceral and Oncologic Surgery, Klinikum Bremen-Mitte, 28177 Bremen, Germany; Gabriele.Boehm@gesundheitnord.de; 6Institute of Ageing and Chronic Disease, University of Liverpool, Liverpool L69 3BX, United Kingdom; kazuhiro.yamamoto@liverpool.ac.uk; 7Digestive Disease Center and Copenhagen Wound Healing Center, Bispebjerg Hospital, University of Copenhagen, 2400 Copenhagen, Denmark

**Keywords:** extracellular matrix, inflammation, cytokines, proteinases, interstitial collagens

## Abstract

Inflammatory processes in the skin augment collagen degradation due to the up-regulation of matrix metalloproteinases (MMPs). The aim of the present project was to study the specific impact of MMP-3 on collagen loss in skin and its interplay with the collagenase MMP-13 under inflammatory conditions mimicked by the addition of the pro-inflammatory cytokine tumor necrosis factor-α (TNF-α). Skin explants from MMP-3 knock-out (KO) mice or from transgenic (TG) mice overexpressing MMP-3 in the skin and their respective wild-type counterparts (WT and WTT) were incubated ex vivo for eight days. The rate of collagen degradation, measured by released hydroxyproline, was reduced (*p* < 0.001) in KO skin explants compared to WT control skin but did not differ (*p* = 0.47) between TG and WTT skin. Treatment with the MMP inhibitor GM6001 reduced hydroxyproline media levels from WT, WTT and TG but not from KO skin explants. TNF-α increased collagen degradation in the WT group (*p* = 0.0001) only. More of the active form of MMP-13 was observed in the three MMP-3 expressing groups (co-incubation with receptor-associated protein stabilized MMP-13 subforms and enhanced detection in the media). In summary, the innate level of MMP-3 seems responsible for the accelerated loss of cutaneous collagen under inflammatory conditions, possibly via MMP-13 in mice.

## 1. Introduction

More than half a century ago, Gross and Lapière discovered collagenase-1, the first matrix metalloproteinase (MMP) [1]. Since then, research on MMP has yielded vast information on these zinc-dependent endopeptidases [2,3]. Expanding knowledge on MMP function and its impact on physiological and pathological processes has increased the interest in MMP substrates and has fueled further research [4]. Stromelysin-1 (MMP-3) belongs to the MMP family and is induced during development [5], wound repair [6], inflammation [7,8] and cancer [9]. Apart from cleaving extracellular matrix proteins, e.g., E-cadherin, laminins and type IV collagen, MMP-3 activates cytokines, growth factors [3,4] and other MMP members, e.g., the collagenases MMP-1, MMP-8 and MMP-13 and gelatinase B (MMP-9) [3]. In contrast to the collagenases MMP-1, MMP-8 and MMP-13, which are the main effectors of type I collagen degradation during wound repair, MMP-3 is incapable of cleaving native type I collagen. MMP-3 has been implicated in many inflammatory diseases, e.g., ultraviolet-B irradiation and photoaging [10], arthritis [11], after lung injury [8], vascular disease [12] and intestinal inflammation [13]. MMP-3-deficient mice show diminished inflammatory responses to various stimuli [14,15] and reduced cutaneous wound contraction [16]. Pro-inflammatory tumor necrosis factor-α (TNF-α) is an important mediator of inflammatory processes. TNF-α induces a variety of MMPs, e.g., MMP-1, MMP-3, MMP-8, MMP-9 and MMP-13 [4]. Apart from TNF-α converting enzyme (TACE) [17], MMP-3 was shown to activate TNF-α [18,19], yielding a 17 kDa peptide [20]. Activated TNF-α induces the downstream expression and secretion of MMP-3 [21,22]. In our previous study, human skin explant cultures challenged with TNF-α reacted with a concomitant and increased MMP-1, MMP-3 and collagen degradation. In this human model, we hypothesized that the increased collagen degradation was due to the activation of the collagenase MMP-1 by MMP-3 [23].

The primary aim of this project is to study the lack of MMP-3 and the effect of excessive MMP-3 on collagen degradation in skin. For this purpose, we incubate skin explants from MMP-3 knock-out [24] and MMP-3 transgenic mice [25]. Under the applied culture conditions, collagen catabolism dominates and the synthesis of new interstitial collagens is minimal [23]. MMP-1 is not expressed in mice but MMP-13 is thought to be the main collagenase and functional homologue of human MMP-1 [26]. Therefore, we study the interplay between MMP-3 and MMP-13. To enhance the extracellular detection of MMP-13, its endocytosis is blocked by adding the receptor-associated protein (RAP). Finally, the responsiveness of MMP-3-deficient or MMP-3-overexpressing murine skin to an inflammatory stimulus provided by TNF-α is studied. The broad-spectrum MMP inhibitor GM6001 is included in some experiments to elucidate the origin of collagen-degrading enzymes. 

## 2. Results

### 2.1. MMP-3 Expression in the Skin of the Four Murine Genotypes

First, we quantified the MMP-3 mRNA (Figure 1A) and protein (Figure 1B) in the native skin of the four groups of mice. MMP-3 mRNA and protein were undetectable in the skin of MMP-3-deficient mice (KO). MMP-3 mRNA (*p* = 0.016) and protein levels were higher in the skin of transgenic (TG) animals compared with their wild-type (WTT) controls. WT and WTT contained similar MMP-3 mRNA and protein (approximately 0.2–0.3 ng) levels.

### 2.2. Collagen Degradation in Incubated Skin Explants of the Four Murine Genotypes

We measured hydroxyproline in the media as an indicator of cleaved collagen molecules in the skin. MMP-3-deficient (KO) skin explants (BL10 genetic background) released a total amount of 90 µg hydroxyproline into the media over the eight-day incubation period, WT (BL10) skin 188 µg, WTT (FVB) skin 304 µg and MMP-3-overexpressing TG (FVB) skin 285 µg hydroxyproline (Table 1). Notably, the accumulated hydroxyproline release was higher in FVB WTT (304 µg) versus BL10 WT (188 µg) mice. Regression analysis was carried out to assess the rate of collagen degradation over the eight-day culture period and showed significant differences in the total hydroxyproline released between KO and WT (*p* < 0.001) skin (Table 1 and Figure 2A) but not between WTT and TG (*p* = 0.47) skin (Table 1 and Figure 2B). 

Functional MMP-3 activity was demonstrated by casein zymography in a conditioned medium from skin explants of WT mice but not from skin explants of KO mice (Figure 3A). Tissue extracts of incubated skin explants produced the two characteristic and specific ¾ and ¼ type I collagen fragments by the action of tissue collagenases. TIMP-1 blocked this endogenous collagenolytic activity. Furthermore, incubation with the organomercurial APMA partly compensated for the lack of MMP-3 (Figure 3B). Consequently, we next wanted to identify the collagenase(s) responsible for the observed collagenolytic activity.

### 2.3. MMP-13 in MMP-3-Deficient and MMP-3-Overexpressing Conditions

Only true tissue collagenases are capable of cleaving native type I collagen. In rodents, these are MMP-1a [27], MMP-8 and MMP-13 [26]. MMP-3 indirectly contributes to collagenolysis via the activation of collagenases [28]. Here, neither MMP-1a nor MMP-8 was detected by western blot analysis. The main collagenase responsible for tissue homeostasis in rodents is also claimed to be MMP-13 [26]. Therefore, we focused on MMP-13 by determining the mRNA and protein levels in native and incubated skin.

In native skin, MMP-13 mRNA levels were similar in the KO, WT and WTT groups, while MMP-13 transcripts were undetectable in TG skin (Figure 4A). After eight days of incubation, MMP-13 mRNA levels differed among the groups with more (*p* = 0.03) MMP-13 mRNA present in KO skin compared with WTT skin (Figure 4B).

The MMP-13 protein in conditioned media was analyzed by western blot analysis (Figure 4C). After four days of culture, bands corresponding to latent (57 kDa) and active (48 kDa) MMP-13 were fainter in KO compared with WT. The catalytic domain of MMP-13, represented by the band at approximately 29 kDa, was prominent in WT but not in KO. Overall, the expression of MMP-13 was higher in WT compared to KO. In WTT, only faint bands of latent MMP-13 and of the catalytic domain fragment at 29 kDa were found. In contrast, TG-conditioned media presented prominent bands at the positions of latent and active MMP-13. Because MMP-13 detection can be jeopardized by rapid intracellular uptake via the LRP-1 receptor [29,30], the LRP-1 receptor antagonist RAP was used to inhibit MMP-13 endocytosis [31]. The presence of RAP during incubation yielded similar MMP-13 bands but fewer intermediate MMP-13 forms and more of the active MMP-13 compared to the controls (no RAP). After six and eight days of culture, no or minimal amounts of MMP-13 were detected in conditioned media.

### 2.4. Effect of Pro-Inflammatory TNF-α on MMP-3 Tissue Levels, MMP-13 Secretion and Collagen Degradation

We next sought to investigate MMP-3 expression under an inflammatory stimulus provided by TNF-α. Expectedly, TNF-α addition increased the presence of latent and active MMP-3 in WT skin but not in KO skin (Figure 5). 

Moreover, TNF-α increased the secretion and activation of latent MMP-13 in the conditioned media of WT and KO after four days of culture with more fragments and less latent MMP-13 in WT and more latent MMP-13 in KO (Figure 6A). In contrast to the effect of TNF-α on skin from these animals with BL10 background, only mild effects were observed in the mice with FVB background (WTT and TG). TNF-α treatment increased the fragmentation of MMP-13 from incubated WTT skin explants and active MMP-13 at 48 kDa from incubated TG skin explants (Figure 6B). 

The incubation of WT skin explants with TNF-α was translated into increased (*p* = 0.0001) collagen degradation compared with the control-treated WT skin explants over the eight-day experimental period (Figure 2A). In contrast, the TNF-α addition to KO skin explants did not increase (*p* = 0.06) collagen degradation further compared with the control. No significant differences in collagen degradation were found between TNF-α-treated WTT and TG skin explants (Figure 2B). 

### 2.5. Effect of Pro-Inflammatory TNF-α on MMP-2 Secretion in MMP-3-Deficient and -Overexpressing Conditions

Apart from MMP-3 [32], MMP-2 plays an auxiliary role in activating latent MMP-13 with the conversion of the intermediate form into the fully activated form of MMP-13 [33]. In the day eight-conditioned media, no significant differences were found among the four groups with or without TNF-α addition, although the FVB (WTT and TG) mice tended to secrete less MMP-2 compared with the BL10 (KO and WT) mice.

## 3. Discussion

Pro-inflammatory MMP-3 and TNF-α are associated with many diseases that are characterized by massive tissue destruction and collagen degradation. Previously, we showed that TNF-α promoted collagenolysis in human skin via the activation of MMP-1, possibly mediated by MMP-3 [23]. Furthermore, earlier studies indicated that MMP-3 increases collagenase activity in cultured murine bone and cartilage [34,35]. To investigate if MMP-3 per se has a role in collagen catabolism, we incubated skin from MMP-3-deficient (KO) and MMP-3-overexpressing mice (TG) and their respective wild-type counterparts (WT and WTT). We found massively reduced collagen degradation and fragmentation in MMP-3-deficient skin, which could possibly be ascribed to the reduced activation of the collagenase MMP-13. 

The observed effect on collagen degradation could entirely be ascribed to the action of MMP-3 because addition of the non-selective MMP inhibitor GM6001 to the WT skin cultures did not decrease collagen loss more than that of the KO group. It should be emphasized that our sample size was small and therefore the risk of making a type II error is apparent.

We isolated proteinases from the incubated murine KO skin that produced the two ¼ and ¾ fragments of native type I collagen triple helix. The extracted proteinases could be activated by APMA and completely blocked by TIMP-1. This implied that one or more collagenases degraded the interstitial collagens of the skin and that the lack of collagenase activation appeared to be an underlying mechanism for reduced collagenolysis in MMP-3 deficiency. Corroborating these data are the findings by van Meurs et al. [35], who observed the specific single cleavage site in type II collagen in cultured cartilage from wild-type mice but not from MMP-3-deficient mice. 

MMP-13 is thought to be the most powerful collagenase under normal and pathologic conditions [36]. The two other rodent collagenases, MMP-1a and MMP-8, were not expressed in significant amounts in our skin explant model. Consequently, we focused on the presence and activation state of MMP-13. We could show that MMP-3-deficient skin expressed similar amounts of MMP-13 mRNA compared to controls. The detection of the secreted MMP-13 protein is difficult due to its high affinity binding to LRP [29,30]. Hence, we added the LRP-1 receptor agonist RAP [31] to block MMP-13 endocytosis. Despite this manipulation, similar amounts of MMP-13 were present in the conditioned media of KO and WT samples treated with RAP. Nevertheless, differences in the MMP-13 levels were unlikely the cause for differential collagenolysis seen in MMP-3-deficient conditions. On the other hand, activated MMP-13 molecular species were reduced in KO skin in comparison with the wild-type counterpart. Apart from MMP-3, trypsin, plasmin, cathepsin B, MMP-2, MMP-14 and possibly MMP-8 can activate latent MMP-13 [32,33]. Here, MMP-2 was most likely not involved. Despite of multiple possibilities for MMP-13 activation, MMP-3 seemed to be the main activator of MMP-13 in our ex vivo skin model and the absence of the activation function of MMP-3 was not compensated for by other proteinases. 

Besides being capable of activating proMMP-13, MMP-3 can activate growth factors and cytokines, notably proIL-1β. IL-1β in turn influences MMP-3 and MMP-13 expression and secretion [37,38].

Having established that collagen loss is severely reduced under MMP-3-deficient conditions and that the activation, rather than the concentration, of MMP-13 decreases collagen in skin, we next investigated the effect of pro-inflammatory TNF-α on MMP-13 expression and collagen loss. TNF-α is known to induce the expression of MMP-3 [23,39], MMP-9 [18,40] and MMP-13 [41]. TNF-α induced MMP-3 in WT but not in KO skin explants. Moreover, TNF-α augmented the activation of MMP-13 in the WT skin explants, presumably via MMP-3 induction. The increased MMP-13 activation by TNF-α would explain the increased collagen degradation and fragmentation in WT skin explants not found in the KO group. One can speculate about the impact of immune cells under in vivo conditions. Lauridsen et al. [42] reported higher levels of MMP-3 when pericytes were activated with TNF-α and co-cultured with neutrophils. Our results may well explain the pronounced collagen degradation and tissue destruction found in highly inflammatory and chronic progressing diseases. 

The dependence of collagen degradation on the genetic background of the mice (BL10 versus FVB) was intriguing. Collagenolysis was not increased in the skin of the FVB mice overexpressing MMP-3. If we assume that the MMP-13 activity is the rate-limiting factor for collagenolysis in our model, then the content of MMP-13 is critical. MMP-13 levels were seemingly higher in MMP-3-overexpressed skin. This may indicate that the activity of MMP-3 was not increased in the MMP-3 transgenic group. Many proteinases activate proMMP-3 [28], for example, mast cell-derived proteinases [43].

Wild-type skin from BL10 mice responded with a massive increase of collagen degradation, MMP secretion and activation when TNF-α was added to cultures, whereas FVB murine skin did not respond to the TNF-α challenge. A similar finding was reported by Martin et al. [44] using a breast cancer model in C57Bl/6 and FVB/N mice. MMP-mediated effects on tumor angiogenesis were only found in C57BL/6 mice, with no effect in the FVB/N strain. Therefore, the choice of experimental animals with the appropriate genetic background is pivotal with respect to the interpretation of results and translation to the human situation. 

## 4. Materials and Methods

### 4.1. Animals

MMP-3-deficient mice and the corresponding wild-type mice were donated by Dr. Gary Rosenberg, University of New Mexico, Health Science Center, Albuquerque, NM, USA [24]. The animals were crossed back to a homogeneous background of BL10 (C57BL10.RIIIH2r-Mmp3tm1Mol/Ztm) mice for 10 generations. Skin from four-month-old or six-month-old homozygous BL10 MMP-3 knock-out (KO) and BL10 wild-type (WT) mice was used in this study. 

Mice overexpressing MMP-3 under the keratin-5 promoter were first described by Dr. Lisa McCawley [25]. This overexpression is confined mainly to the keratinocytes of the skin. The sperm of MMP-3-overexpressing mice was used to inseminate mice of the FVB genotype. The offspring were mated to mice with a homozygous background. Backcrossing was performed for 10 generations. Skin from three-month-old FVB MMP-3-overexpressing, transgenic (TG) and FVB wild-type (WTT) mice were used. 

The animals were kept in Makrolon^®^ Type 2L cages at 21 °C, with a relative humidity of 55% ± 5% and a light cycle of 10 h/14 h. Sterilized commercial softwood granulate bedding was used (Lignocel, Altromin, Lage, Germany). Autoclaved commercial pellets (Altromin 1314, Altromin, Lage, Germany) and autoclaved water were provided ad libitum. The microbiological status was examined as recommended by FELASA (Federation of European Laboratory Animal Science Associations) and the absence of listed microorganisms was confirmed [45]. We did not test for the presence of *Rodentibacter sp., Helicobacter spp., Staphylococcus aureus* or *Klebsiella oxytoca*.

The experiments were approved by the local Institutional Animal Care and Research Advisory Committee (522-27-11/02-00/118, 10 September 2013).

### 4.2. Experimental Model and Treatment Groups

After decapitation, the dorsal skin from the male mice of all genotypes was shaved, disinfected and excised under sterile conditions. The skin samples (approximately 5 cm × 5 cm) were prepared within 36 h after harvesting. The skin explants were excised with a sterile 8 mm or a sterile 4 mm trephine and incubated submerged in a serum-free culture medium in 24-well culture plates (Nunc, Roskilde, Denmark) at 37 °C in a humidified atmosphere of 5% CO_2_/air in a keratinocyte growth medium (KGM)-2 containing 6 mmol/L glucose, 50 ng/mL amphotericin-B, 100 µg/mL penicillin, 100 U/mL streptomycin and supplemented with 1.4 mM CaCl_2_ [23]. 

For each animal genotype, i.e., KO, WT, WTT or TG (Table 2), 8 mm skin explants were incubated in 1.0 mL medium in the absence (control) or presence of rmTNF-α (Sigma-Aldrich, St. Louis, MO, USA) or the broad-spectrum hydroxamate MMP inhibitor GM6001 (Merck-Millipore, Molsheim, France).

Because MMP-13 is rapidly endocytosed intracellularly, extracellular detection of MMP-13 can be difficult. The internalization of MMP-13 is mediated by LRP-1 [29,30]. RAP is a ligand-binding antagonist for LRP-1 and inhibits proteinase endocytosis [31]. Hence, we used RAP in a second series of experiments to facilitate extracellular MMP-13 detection by blocking rapid internalization. Skin explants (4 mm) were incubated in 0.25 mL medium in the absence (control) or presence of the indicated treatments in Table 3. 

Conditioned media were harvested and replaced by a fresh medium every second day. After eight days of incubation, the experiments were terminated, media were harvested, and skin explants were kept at −80 °C until analysis for MMP-3 and MMP-13 mRNA, and MMP-3 protein. Conditioned media were stored at −20 °C until analysis for hydroxyproline and MMP-13 protein.

### 4.3. Analyses

#### 4.3.1. Hydroxyproline Assay

Hydroxyproline levels in the conditioned media from days two, four, six and eight were measured colorimetrically [23].

#### 4.3.2. MMP-3 and MMP-13 mRNA Determined by qPCR

RNA was isolated from skin explants (approximately 30 mg) using TRIzol reagent (Thermo Fisher Scientific, Schwerte, Germany). Total RNA (approximately 100 ng) were reverse-transcribed into cDNA and subsequently analyzed for 40 cycles in qPCR. The qPCR was conducted on a 7500 Fast Real Time PCR system (Thermo Fisher Scientific, Schwerte, Germany) with TaqMan probes for murine MMP-3 (Mm00440295_m1) and murine MMP-13 (Mm00439491_m1). Murine β-actin (Mm02619580_g1) and murine peptidylprolyl isomerase A (Mm02342430_g1) were used as housekeeping genes. Each reaction was run in triplicate and the relative quantity of MMP-3 or MMP-13 was determined using the ΔΔCT method [46]. 

#### 4.3.3. Casein and Gelatin Zymography

Casein zymography (β-casein, Sigma-Aldrich, St. Louis, MO, USA) was performed as described elsewhere [23]. Mark12™ unstained protein standard (LC5677, Thermo Fisher Scientific, Schwerte, Germany) was run in a parallel lane. After electrophoresis, the gels were incubated in 50 mM Tris-HCl (pH 7.5) containing 10 mM CaCl_2_, 1 µM ZnCl_2_ and 0.1% Triton X-100 with or without the selective MMP-3 inhibitor UK370106 (R&D Systems, Minneapolis, MN, USA) for 72 h at 37 °C.

Gelatin zymography was used to analyze the MMP-2 content in 4× concentrated (Amicon Ultra-0.5 filter, Merck-Millipore, Darmstadt, Germany) conditioned media [23]. rmMMP-2 (924-MP, R&D Systems, Minneapolis, MN, USA) was run at 2 ng in parallel to estimate the MMP-2 content of the samples by the image analysis of digitized gels [47].

#### 4.3.4. Collagenase Activity

Tissue extracts of skin explants that had been incubated for eight days were prepared together with 1 mM Pefabloc SC (Roche Diagnostics GmbH, Mannheim, Germany) and concentrated 4x by Amicon Ultra-0.5 [23]. The collagenolytic activity was determined by incubating the extracts with native (trypsin-resistant) type I collagen from bovine skin for 240 h at 22 °C, as described elsewhere [23]. rhTIMP-1 (R&D Systems, Minneapolis, MN, USA) was added to one set of reaction vials to a final concentration of 200 nM and APMA (Sigma-Aldrich, St. Louis, MO, USA) to another set of reaction vials to a final concentration of 1 mM. TIMP-1 or APMA were present during the entire 240 h incubation period.

#### 4.3.5. Western Blot Analyses

Native and incubated skin explants were homogenized (Ultra-Turrax^®^ T25 Basic, IKA Werke GmbH, Staufen, Germany) in RIPA (radioimmunoprecipitation assay) buffer or modified RIPA buffer at pH 7.4 (5 µL/mg tissue) containing 0.1 M Tris-HCl, 0.15 M NaCl, 1% Triton X-100, 0.1% SDS (sodium dodecyl sulfate) [48] and EDTA (ethylendiamintetraacetate)-free proteinase inhibitor cocktail and 1 µM pepstatin A. Homogenates were then centrifuged at 12,000× g for 10 min, and the supernatants were stored at −20 °C until analysis. Protein concentration was determined using the Pierce BCA (bicinchoninic acid) Protein Assay kit (Thermo Fisher Scientific, Schwerte, Germany).

Tissue extracts, conditioned media, molecular weight markers (RPN800E, Sigma-Aldrich, St. Louis, MO, USA) and rmMMP-3 (RPA101Mu01, Cloud-Clone Corporation, Katy, TX, USA) were electrophoresed on 10% SDS-PAGE gel under reducing conditions and electrotransferred for seven min using iBlot apparatus (Invitrogen, Carlsbad, CA, USA) onto a polyvinylidene fluoride (PVDF) membrane (Immobilon^®^, Merck-Millipore, Darmstadt, Germany). Membranes were blocked in 5% fat-free milk (Bio-Rad Laboratories, Hercules, CA, USA) for 1 h at room temperature and incubated with primary antibodies for 18 h at 4 °C (rabbit polyclonal anti-mouse MMP-1a antibody (1:200 dilution; 250750, Abbiotec, San Diego, CA, USA), anti-mouse MMP-3 peptide antibody [49,50], rabbit polyclonal anti-MMP-8 (1:1000 dilution; ABT38, Merck-Millipore, Darmstadt, Germany), rabbit polyclonal anti-MMP-13 antibody (1:250 dilution; ab39012, Abcam, Cambridge, UK), mouse monoclonal anti-β-actin antibody (1:40,000 dilution; A5441, Sigma-Aldrich, St. Louis, MO, USA), rabbit polyclonal anti-β-actin antibody (1:1000 dilution; #4967, Cell Signaling Technology, Leiden, The Netherlands) or rabbit monoclonal anti-GAPDH antibody (1:1000 dilution; #2118, Cell Signaling Technology, Leiden, The Netherlands)). Then, membranes were incubated for 1 h at room temperature with IgG-horseradish peroxidase secondary antibodies (Jackson ImmunoResearch, Ely, UK), and a specifically bound antibody was detected using an immunodetection kit (Amersham ECL Prime Western Blotting Detection Reagent, GE Healthcare Life Sciences, Amersham, UK), and then exposed to X-ray film (Amersham Hyperfilm ECL, GE Healthcare Life Sciences, Amersham, UK).

For the analysis of MMP-13 in conditioned media by western blot, the loading of samples was normalized from Coomassie blue-stained 10% SDS-PAGE gels because the BCA assay was not sensitive enough for these samples. On each gel, the samples were run with a reference sample composed of conditioned media from all control-treated samples (50 µL of each sample were mixed) in each specific experiment. The density of a common band at approximately 70 kDa was determined by ImageJ™ and used for the normalization of protein loading (Appendix A).

### 4.4. Statistics

The Shapiro–Wilk test was used to assess for normality. A Student’s t-test was used for the statistical analysis of results with normal distribution, and otherwise the Wilcoxon Rank Sum test and the Mann–Whitney U test using GraphPad Prism 8.0.2 software (GraphPad Software Inc., San Diego, CA, USA) were applied. For the comparison of hydroxyproline release over time, the slopes for two independent samples were compared after regression analysis. The Bonferroni correction was used in the cases of multiple comparisons. *p* < 0.05 was considered significant.

## 5. Conclusions

Collagen degradation in murine skin is MMP-3 and MMP-13 dependent, with MMP-3 being a potent activator of MMP-13. The pro-inflammatory cytokine TNF-α enhanced collagenolysis via the up-regulation of MMP-3 and increased the activation of MMP-13. Co-targeting MMP-3 in inflammatory diseases seems an appropriate measure to enhance the anti-inflammatory impact of clinically used drugs.

## Figures and Tables

**Figure 1 ijms-20-05234-f001:**
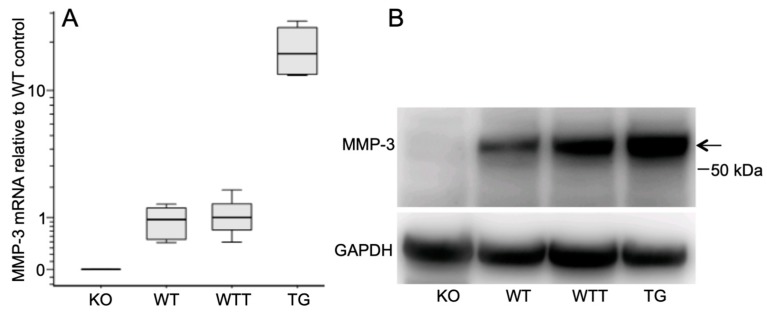
MMP-3 expression in native mouse skin. (**A**) MMP-3 mRNA levels determined by qPCR analysis. Boxes represent the 25th–75th percentile, whiskers the 5th–95th percentile and the horizontal lines within the boxes indicate the median values. (**B**) Representative western blot of six independent experiments of MMP-3 protein expression. Loading was normalized to the total protein (30 µg/lane) of the tissue extracts. GAPDH indicates the loading/transfer of proteins to the PVDF membrane. The arrow indicates the latent MMP-3 at 56 kDa. The position of the 50 kDa molecular weight marker is indicated to the right. KO (BL10), *n* = 5; WT (BL10), *n* = 5; WTT (FVB), *n* = 3; TG (FVB), *n* = 4.

**Figure 2 ijms-20-05234-f002:**
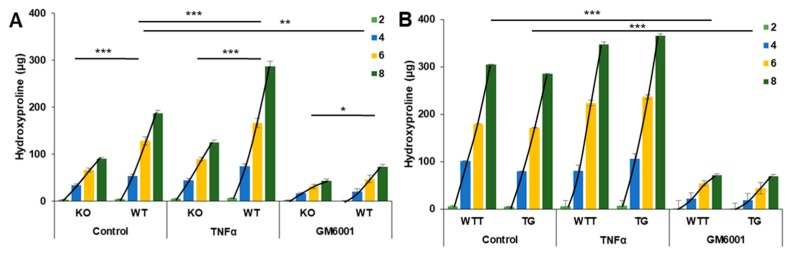
The effect of the MMP-3 genotype (control), TNF-α treatment and GM6001 treatment on collagen degradation in skin measured by the release of hydroxyproline-containing peptides into media from KO and WT (**A**), and WTT and TG (**B**) skin explants incubated over eight days. The timely progression of hydroxyproline release was assessed by regression analysis of the slopes. Mean ± SEM (bars). * *p* < 0.05, ** *p* < 0.01, *** *p* < 0.001. The number of animals (*n*) used for theses analyses is given in Table 2. TNF-α (10 ng/mL). GM6001, broad-spectrum MMP inhibitor (10 µM). KO, MMP-3 knock-out; WT, wild-type control to KO; WTT, wild-type to transgenic mice (TG); TG, MMP-3 overexpression in skin. Light green bars, day 2; blue bars, day 4; yellow bars, day 6; dark green bars, day 8.

**Figure 3 ijms-20-05234-f003:**
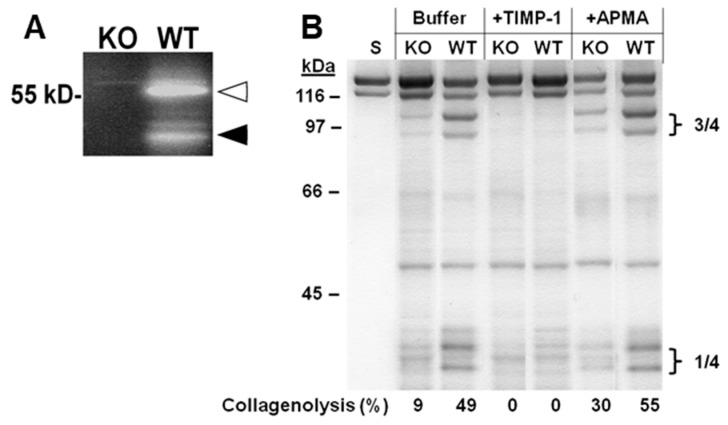
MMP-3 (**A**) and collagenase (**B**) activities. (**A**) Casein zymography of conditioned media collected after eight days of incubation of KO and WT skin explants. Addition of the selective MMP-3 inhibitor UK370106 (1 µM) during incubation of the casein containing SDS-PAGE gels abolished the lysis bands at the arrowheads observed in the control WT media. Open arrowhead, latent MMP-3; solid arrowhead, active MMP-3. (**B**) Type I collagenolytic activity of tissue extracts of skin explants incubated for eight days. Collagenase activity of the extracts was determined in the absence (buffer) or in the presence of rhTIMP-1 (200 nM) or APMA (1 mM). KO, MMP-3 knock-out; WT, wild-type control to KO.

**Figure 4 ijms-20-05234-f004:**
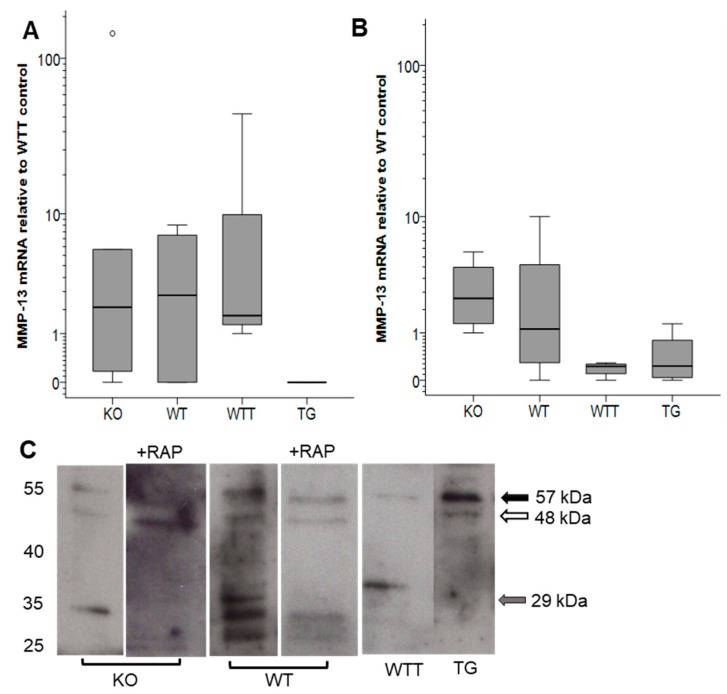
MMP-13 mRNA (**A**,**B**) and protein expression (**C**) in MMP-3 deficiency and overexpression. (**A**,**B**) MMP-13 mRNA expression in (**A**) native skin and (**B**) in incubated skin after eight days of culture. Boxes represent the 25th–75th percentile, whiskers the 5th–95th percentile and the horizontal lines within the boxes indicate the median values. (**C**) MMP-13 protein in conditioned media after four days of the incubation of skin explants analyzed by western blot. Molecular weights in kDa are indicated to the left. The black arrow to the right indicates the position of latent MMP-13 (57 kDa), the white arrow indicates the active MMP-13 (48 kDa) and the grey arrow to the right indicates the fragment containing the catalytic domain (29 kDa). The band below the 29 kDa band represents the C-terminal domain of MMP-13 [30]. The displayed western blot is representative of four animals in each group. The loading of samples was calculated relative to a pool of day four-conditioned media from all control samples (50 µL of each sample were mixed). The reference pool was run together with the samples on 10% SDS-PAGE gels that were stained with Coomassie blue. The density of the common band at approximately 70 kDa was used for normalization (Appendix A). KO, MMP-3 knock-out; WT, wild-type control to KO; WTT, wild-type to transgenic mice (TG); TG, MMP-3 overexpression in the skin. +RAP, addition of receptor-associated protein (RAP).

**Figure 5 ijms-20-05234-f005:**
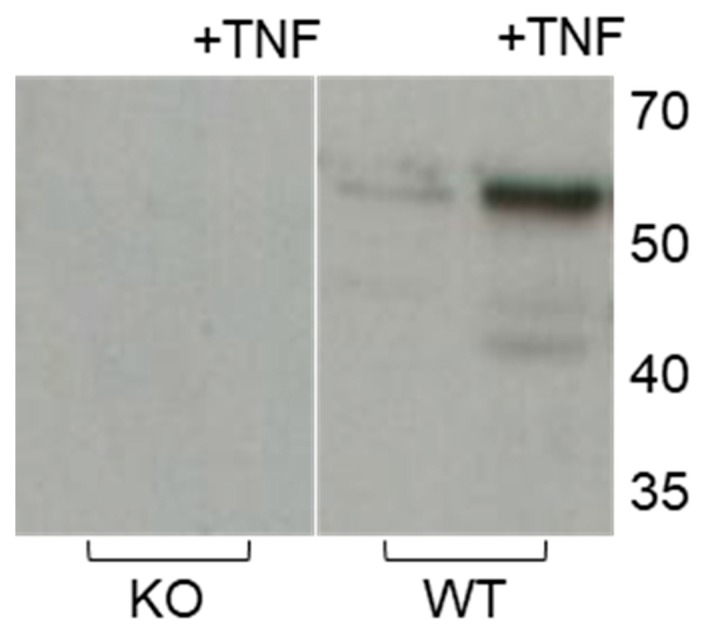
Effect of TNF-α on MMP-3 tissue levels in KO and WT skin after eight days of incubation in the absence or presence of TNF-α (+TNF) analyzed by western blot. Loading was normalized to the total protein (10 µg/lane) of the tissue extracts. Molecular weights in kDa are indicated to the right. Positions of protein markers in kDa are to the right. KO (*n* = 4), MMP-3 knock-out skin; WT (*n* = 4), wild-type control to KO skin.

**Figure 6 ijms-20-05234-f006:**
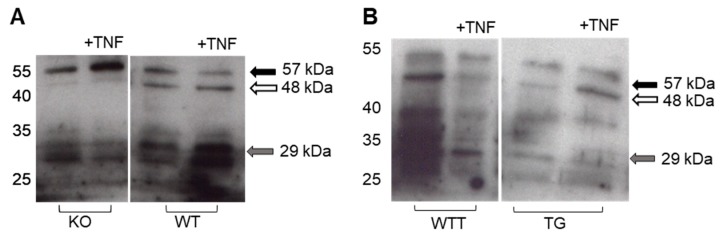
Effect of TNF-α on MMP-13 levels in conditioned media of KO and WT (**A**), and WTT and TG (**B**) skin explants incubated without or with 10 ng/mL TNF-α (+TNF) for four days and analyzed by western blot. The black arrow indicates the position of latent MMP-13 (57 kDa), the white arrow indicates the active MMP-13 (48 kDa) and the grey arrow the fragment containing the catalytic domain (29 kDa). Loading of samples was calculated relative to a pool of day four-conditioned media from all (*n* = 17) control samples (50 µl of each sample were mixed). The reference pool was run together with the samples on 10% SDS-PAGE gels that were stained with Coomassie blue. The density of the common band at approximately 70 kDa was used for normalization (Appendix A). The positions of protein markers in kDa are indicated to the left. KO (*n* = 5), MMP-3 knock-out; WT (*n* = 5), wild-type control to KO; WTT (*n* = 3), wild-type to transgenic (TG); TG (*n* = 4), MMP-3 overexpression in skin.

**Table 1 ijms-20-05234-t001:** Hydroxyproline (µg) in conditioned media over eight days of culture (mean ± SEM).

Day	KO (*n* = 10)	WT (*n* = 10)	WTT (*n* = 9)	TG (*n* = 10)
0	0	0	0	0
2	3.9 ± 0.2	4.3 ± 0.5	6.8 ± 1.7	5.0 ± 1.0
4	29.9 ± 3.5	49.3 ± 4.7	94.3 ± 12.0	74.4 ± 10.5
6	32.4 ± 3.9	74.9 ± 8.4	79.0 ± 5.7	92.4 ± 11.2
8	24.1 ± 3.6	59.0 ± 5.5	124.4 ± 16.7	113.2 ± 11.8
Accumulated	90.3	187.5	304.5	285.0
Regression analysis				
Slope	3.16	9.49	16.9	17.1
*p*	<0.001		0.47	

**Table 2 ijms-20-05234-t002:** Experimental groups.

Group	Control	TNF-α ^1^	GM6001 ^2^
KO	*n* = 10	*n* = 10	*n* = 5
WT	*n* = 10	*n* = 10	*n* = 5
WTT	*n* = 9	*n* = 9	*n* = 4
TG	*n* = 10	*n* = 10	*n* = 5

^1^ 10 ng/mL. ^2^ 10 µM.

**Table 3 ijms-20-05234-t003:** Experimental setting for the RAP (receptor-associated protein) experiment.

Group	Control		TNF-α ^1^		GM6001 ^2^	
	0	+RAP ^3^	0	+RAP ^3^	0	+RAP ^3^
KO	*n* = 4	*n* = 4	*n* = 4	*n* = 4	*n* = 4	*n* = 4
WT	*n* = 4	*n* = 4	*n* = 4	*n* = 4	*n* = 4	*n* = 4
WTT	*n* = 4	*n* = 4	*n* = 4	*n* = 4	*n* = 4	*n* = 4
TG	*n* = 4	*n* = 4	*n* = 4	*n* = 4	*n* = 4	*n* = 4

^1^ 10 ng/mL. ^2^ 10 µM. ^3^ 250 nM.

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
