# Peer review of "Matrix Metalloproteinase-3 is Key Effector of TNF-α-Induced Collagen Degradation in Skin"

_ijms, 2019, doi:10.3390/ijms20205234_

Round 1

Reviewer 1 Report

In the revised form the authors have addressed all my questions and recommendations. I do not have further concerns.

Reviewer 2 Report

This manuscript has been extensively revised, presumably in response to a previous review. The main conclusion is that MMP-3 mediates MMP-13 activation in response to TNF-alpha in a mouse model. TNF-alpha upregulates MMP-3, which in turn activates MMP-13. This will be of considerable interest to the medical community, especially dermatologists and the dental community, especially periodontists. I do not think that further editing is justified.

After extensive editing, this manuscript is acceptable for publication.

This manuscript is a resubmission of an earlier submission. The following is a list of the peer review reports and author responses from that submission.

Round 1

Reviewer 1 Report

The paper by Mirastschijski reports on experiments using mouse models to study the effect of MMP-3 induction on the activation of down-stream MMP-13 which has collagenolytic activity. To approach this question MMP-3 k.o. as well as overexpressing (TG) animals were used.

Major points:

Fig. 1 The authors show that the respective skin tissues produce no (k.o.) moderate (WT and WTT) and strongly increased MMP-3 mRNA (TG). For k.o. and WT also western blots as well as zymograms showing respective MMP-3 protein and activity levels are presented. Why not for WTT and TG skin?   Why include TG data at all, if the basics have not been analyzed or presented?

Table. 1 The collagenolytic activity as measure by hydroxyproline release of k.o. and WT extracts show a higher activity in WT extracts. However, overexpression of MMP-3 (TG vs WTT) does not lead to an increase. Why is that? Therefore, the experiments mentioned above are so important. Is only MMP-3 mRNA increased in TG mice without respective increased protein synthesis/secretion? Another serious problem seems to be that the transgenic animals were produced in different mouse strains and therefore even the two WT lines differ quite substantially in some of the parameters studied. That makes the whole experimental set-up questionable.

Fig. 2 The authors analyze MMP-13 mRNA expression and find that in skin extracts TG does not express MMP-13 whereas its WTT counterpart does. No explanation provided. Also after 8 d in culture the MMP-13 expression in WTT is lost, in contrast to both k.o. and WT extracts. On the other hand MMP-13 protein is differentially expressed in WTT and TG with an increase in TG. This does not fit to the mRNA data. Also the blot shown in Fig. 2C is not acceptable since each lane is cut from a different exposure as can be assumed by the different background staining. What loading control was used?

Fig. 3 Whereas Fig.2 shows a differential expression of MMP-13 in WTT and TG, in Fig. 3 the expression is comparable. Again no loading controls are provided and patched western blots. And why does TNF induce MMP-13 protein expression in k.o. but not in WT tissues? 

Author Response

We thank the reviewer for the very useful comments. Below follow our responses.

Fig. 1 The authors show that the respective skin tissues produce no (k.o.) moderate (WT and WTT) and strongly increased MMP-3 mRNA (TG). For k.o. and WT also western blots as well as zymograms showing respective MMP-3 protein and activity levels are presented. Why not for WTT and TG skin?   Why include TG data at all, if the basics have not been analyzed or presented?  

This is an excellent comment and suggestion. Unfortunately we have no conditioned media left for Western blot analysis, ELISA and casein zymography. We have analyzed tissue extracts of skin biopsies (uncultured skin) of the four genotypes by Western blot but the results were inconsistent. 

Table. 1 The collagenolytic activity as measure by hydroxyproline release of k.o. and WT extracts show a higher activity in WT extracts. However, overexpression of MMP-3 (TG vs WTT) does not lead to an increase. Why is that? Therefore, the experiments mentioned above are so important. Is only MMP-3 mRNA increased in TG mice without respective increased protein synthesis/secretion? Another serious problem seems to be that the transgenic animals were produced in different mouse strains and therefore even the two WT lines differ quite substantially in some of the parameters studied. That makes the whole experimental set-up questionable.  DONE argumentation is located in the discussion.

This is correctly pointed out by the reviewer and we agree that knowledge about the actual MMP-3 levels (especially active MMP-3) would be extremely useful. But those studies were beyond the scope of our project. We have no explanation for the differences between the mouse strains and why overexpression of MMP-3 per se does not lead to an increased collagen degradation in animals with FVB background. We speculate in the Discussion on page 8 (line 245-248) that although more MMP-3 is expressed this may not lead to increased active MMP-3. 

Fig. 2 The authors analyze MMP-13 mRNA expression and find that in skin extracts TG does not express MMP-13 whereas its WTT counterpart does. No explanation provided. Also after 8 d in culture the MMP-13 expression in WTT is lost, in contrast to both k.o. and WT extracts. On the other hand MMP-13 protein is differentially expressed in WTT and TG with an increase in TG. This does not fit to the mRNA data. Also the blot shown in Fig. 2C is not acceptable since each lane is cut from a different exposure as can be assumed by the different background staining. What loading control was used?

We have no explanation for for the lack of correlation between MMP-13 mRNA and protein measurements.  Western blots were configured according to density calculations of Coomassie blue-stained gels. Therefore, an erratic configuration of Western blot gels was the consequence and hence, the cutting of the pictures to provide a logic grouping of samples. The method of loading control is now described in the manuscript on page 11, lines 371-375. "Loading for the Western blot analysis of MMP-13 in conditioned media was calculated from Coomassie blue-stained 10% SDS-PAGE gels. On each gel, the samples were run with a reference sample composed of conditioned media from all control-treated samples (50 µl of each sample was mixed) in each specific experiment. The density of the common ~70 kDa band was determined by ImageJ™ and used for normalization of loading."

Fig. 3 Whereas Fig.2 shows a differential expression of MMP-13 in WTT and TG, in Fig. 3 the expression is comparable. Again no loading controls are provided and patched western blots. And why does TNF induce MMP-13 protein expression in k.o. but not in WT tissues? 

TNF-α induces MMP-13 also in WT with increased amount of activated MMP-13 and more fragmentation products of MMP-13 found at 29 kDa due to higher amounts of MMP-3 compared to KO. Lack of MMP-3 is presumable the reason why less degradation and activation and fragmentation of MMP-13 is seen in KO compared to WT. This argumentation was added to the Discussion section and highlighted in yellow on page 8, lines 247-249. Loading was normalized from the Coomassie blue-stained SDS-PAGE gels (please see our comments above about loading for Western blot). This has been added to the manuscript on page 11, lines 371-375. 

Reviewer 2 Report

The research article entitled “Matrix metalloproteinase-3 is key effector of TNF-α-induced collagen degradation in skin” (ID ijms-551337), investigates the role of MMP-3 in collagen loss of the skin, through ex-vivo cultures of skin explants from MMP-3 knock-out (KO) mice, transgenic (TG) mice overexpressing MMP-3 in the skin and their respective wild-type counterparts (WT and WTT).

They studied both the effect of TNF-α induced inflammatory stimuli, and GM6001-promoted MMP inhibition. The authors observed i) decreased collagen degradation in KO vs. WT control skin; ii) reduced collagen degradation in WT, WTT and TG after treatment with the MMP inhibitor GM6001; iii) increased collagen degradation in the WT group after TNF-α stimulation; iv) increased levels of the active form of MMP-13 in the three MMP-3 expressing groups. They concluded that MMP-3 and MMP-13 are the major players in collagen degradation in murine skin.

The topic is interesting and original. However, several aspects need to be clarified.

Major comments

1.     In figure 1B and 1C are reported the western blotting and casein zymography carried out for KO and WT. The authors studied mRNA levels of MMP-3 for the four groups considered (WT, KO, WTT, TG). The quantification of MMP-3 levels in conditioned media from WTT and TG mice represents an important requirement for correct interpretation of the results. Please, provide information about MMP-3 protein levels in WTT and TG.

2.     In table 1 are reported the levels of hydroxyproline quantified in culture media after 2, 4, 6 and 8 days. However, table 2 is lacking of sample size for each measurement. Please indicate sample size and SD/SEM.

3.     Lines 92-94: The authors wrote “A significant reduction of collagen degradation products was found with GM6001 in WTT and TG control media (WTT, p = 0.002; TG, p = 0.0006, Figure 1F)”. Please, correct (“A significant reduction of collagen degradation products was found with GM6001 in WTT and TG VS. control media (WTT, p = 0.002; TG, p = 0.0006, Figure 1E)”.).

4.     Results referred to Figure 1F lack of replicates and statistical analysis. As it is, the collagenolyis % is referred to a single experiment. Please, provide data about variability (SD/SEM). 

5.      Lines 112-113: The authors described that the number of animals (n) used for these analyses is given in Table 2. However, it is not clear if the sample size for the results obtained for culture media (Figure 1D and E) are referred to n=10 or other. Please, specify it in the figure caption.

6. Assays with rhTIMP-1 (200 nM)/APMA (1 mM) have not been reported in materials and methods. Please, provide information regarding these incubations (time and temperature).

Author Response

We thank the reviewer for the very useful comments. Below follow our responses.

In figure 1B and 1C are reported the western blotting and casein zymography carried out for KO and WT. The authors studied mRNA levels of MMP-3 for the four groups considered (WT, KO, WTT, TG). The quantification of MMP-3 levels in conditioned media from WTT and TG mice represents an important requirement for correct interpretation of the results. Please, provide information about MMP-3 protein levels in WTT and TG.

This is an excellent comment and suggestion and would add valuable data to our manuscript. Unfortunately we have no conditioned media left for ELISA, Western blot analysis or casein zymography. We have analyzed tissue extracts of skin biopsies (uncultured skin) of the four genotypes by Western blot but the results were inconsistent. 

In table 1 are reported the levels of hydroxyproline quantified in culture media after 2, 4, 6 and 8 days. However, table 2 is lacking of sample size for each measurement. Please indicate sample size and SD/SEM.

Sample sizes and SEM have been added to Table 1.

Lines 92-94: The authors wrote “A significant reduction of collagen degradation products was found with GM6001 in WTT and TG control media (WTT, p = 0.002; TG, p = 0.0006, Figure 1F)”. Please, correct (“A significant reduction of collagen degradation products was found with GM6001 in WTT and TG VS.control media (WTT, p = 0.002; TG, p = 0.0006, Figure 1E)”.).

We thank the reviewer for this remark; the text was changed accordingly.

Results referred to Figure 1F lack of replicates and statistical analysis. As it is, the collagenolyis % is referred to a single experiment. Please, provide data about variability (SD/SEM).

This analysis was not intended for quantitative comparison but more a qualitative analysis. Accordingly, we have deleted the sentence: "The collagenolytic activity of KO tissue extracts was substantially reduced compared with wild-type (WT) tissue extracts." to avoid any confusion. The analysis confirmed the presence of a collagenase in both groups.

Lines 112-113: The authors described that the number of animals (n) used for these analyses is given in Table 2. However, it is not clear if the sample size for the results obtained for culture media (Figure 1D and E) are referred to n=10 or other. Please, specify it in the figure caption.

This is very good point. Figure 1D and 1E has been separated into new Figure 2A and B to facilitate the comprehension. Hopefully this change should eliminate any doubts about the number of animals used for the analyses. The citation to Table 2 with the number of animals used is highlighted in yellow.

Assays with rhTIMP-1 (200 nM)/APMA (1 mM) have not been reported in materials and methods. Please, provide information regarding these incubations (time and temperature).

This highlighted information has now been added to the MS on page 10, lines 347-349.

Reviewer 3 Report

This is a straight forward study investigating the effects of MMP3 on MMP13 activity in skin. The data are convincing and conclusions are sound. I have some minor points to raise:

- MMP3 data are shown as mRNA. ELISA would have been nice to see protein data.

- Western blots for MMP13 need improvement. It seems as if the primary antibody [c] is too high, since unspecific bands pop up. Also a WB quantification would be nice. Why is there no zymography done with MMP13 although this is the major read-out of this study?

- Some studies suggest an activation of MMP13 by MMP2 and possibly MMP8. Are these MMPs not expressed? 

Author Response

We thank the reviewer for the complements and comments.

- MMP3 data are shown as mRNA. ELISA would have been nice to see protein data.

This is an excellent comment and suggestion. Unfortunately we have no conditioned media left for ELISA, Western blot analysis and casein zymography. We have analyzed tissue extracts of skin biopsies (uncultured skin) of the four genotypes by Western blot but the results were inconsistent. 

- Western blots for MMP13 need improvement. It seems as if the primary antibody [c] is too high, since unspecific bands pop up. Also a WB quantification would be nice. Why is there no zymography done with MMP13 although this is the major read-out of this study?

This remark is very relevant and we agree with the reviewer. We tried different dilutions of the primary polyclonal antibody and found that a 10 times lower dilution than recommended by to the manufacturer was required to detect MMP-13. The application of an monoclocal MMP-13 antibody (LIPCO IID1) did not yield better results. We have cropped the Western blots to improve the presentation of the MMP-13. Hopefully they are displayed clearer now. We tried to quantitate the Western blots but the results were not useful. Finally, gelatin zymographic analysis did not yield reproducible results for MMP-13. Hence, Western blot analyses were performed.

- Some studies suggest an activation of MMP13 by MMP2 and possibly MMP8. Are these MMPs not expressed? 

The is an excellent remark. We found no significant differences in MMP-2 levels in conditioned media by gelatin zymographic analysis. Furthermore, MMP-8 was not expressed in our skin explants by Western blot analysis. These experiments are now described in the Results, Methods and Discussion sections.

Round 2

Reviewer 1 Report

Author's Notes

We thank the reviewer for the very useful comments. Below follow our responses.

Fig. 1 The authors show that the respective skin tissues produce no (k.o.) moderate (WT and WTT) and strongly increased MMP-3 mRNA (TG). For k.o. and WT also western blots as well as zymograms showing respective MMP-3 protein and activity levels are presented. Why not for WTT and TG skin?   Why include TG data at all, if the basics have not been analyzed or presented?  

This is an excellent comment and suggestion. Unfortunately we have no conditioned media left for Western blot analysis, ELISA and casein zymography. We have analyzed tissue extracts of skin biopsies (uncultured skin) of the four genotypes by Western blot but the results were inconsistent. 

This answer and the removal of the respective western blot is not acceptable. When no conditioned media is „left“, the experiment has to be repeated. And „inconsistent results“ make the whole story unconvincing.

Table. 1 The collagenolytic activity as measure by hydroxyproline release of k.o. and WT extracts show a higher activity in WT extracts. However, overexpression of MMP-3 (TG vs WTT) does not lead to an increase. Why is that? Therefore, the experiments mentioned above are so important. Is only MMP-3 mRNA increased in TG mice without respective increased protein synthesis/secretion? Another serious problem seems to be that the transgenic animals were produced in different mouse strains and therefore even the two WT lines differ quite substantially in some of the parameters studied. That makes the whole experimental set-up questionable.  DONE argumentation is located in the discussion.

This is correctly pointed out by the reviewer and we agree that knowledge about the actual MMP-3 levels (especially active MMP-3) would be extremely useful. But those studies were beyond the scope of our project. We have no explanation for the differences between the mouse strains and why overexpression of MMP-3 per se does not lead to an increased collagen degradation in animals with FVB background. We speculate in the Discussion on page 8 (line 245-248) that although more MMP-3 is expressed this may not lead to increased active MMP-3. 

But those studies should be the scope of your project. Speculation in the disussion section is easy, but cannot replace appropriate experimental data.

Fig. 2 The authors analyze MMP-13 mRNA expression and find that in skin extracts TG does not express MMP-13 whereas its WTT counterpart does. No explanation provided. Also after 8 d in culture the MMP-13 expression in WTT is lost, in contrast to both k.o. and WT extracts. On the other hand MMP-13 protein is differentially expressed in WTT and TG with an increase in TG. This does not fit to the mRNA data. Also the blot shown in Fig. 2C is not acceptable since each lane is cut from a different exposure as can be assumed by the different background staining. What loading control was used?

We have no explanation for for the lack of correlation between MMP-13 mRNA and protein measurements.  Western blots were configured according to density calculations of Coomassie blue-stained gels. Therefore, an erratic configuration of Western blot gels was the consequence and hence, the cutting of the pictures to provide a logic grouping of samples. The method of loading control is now described in the manuscript on page 11, lines 371-375. "Loading for the Western blot analysis of MMP-13 in conditioned media was calculated from Coomassie blue-stained 10% SDS-PAGE gels. On each gel, the samples were run with a reference sample composed of conditioned media from all control-treated samples (50 µl of each sample was mixed) in each specific experiment. The density of the common ~70 kDa band was determined by ImageJ™ and used for normalization of loading."

I don’t think that this way of „normalizing“ gels is good science. The coomassie stained gel and the respective blots are two different things. E.g. the transfer of proteins could be unequal. The loading controls should be done with the uncut blots. And anyway where is the 70 kDa band?

Fig. 3 Whereas Fig.2 shows a differential expression of MMP-13 in WTT and TG, in Fig. 3 the expression is comparable. Again no loading controls are provided and patched western blots. And why does TNF induce MMP-13 protein expression in k.o. but not in WT tissues? 

TNF-α induces MMP-13 also in WT with increased amount of activated MMP-13 and more fragmentation products of MMP-13 found at 29 kDa due to higher amounts of MMP-3 compared to KO. Lack of MMP-3 is presumable the reason why less degradation and activation and fragmentation of MMP-13 is seen in KO compared to WT. This argumentation was added to the Discussion section and highlighted in yellow on page 8, lines 247-249. Loading was normalized from the Coomassie blue-stained SDS-PAGE gels (please see our comments above about loading for Western blot). This has been added to the manuscript on page 11, lines 371-375.